# Agent-Based Approach to Configure Processes in Iran's Banking Service Supply Chain

Negar Jalilian [1] , Seyed Mahmoud Zanjirchi [1,*], Alireza Naser Sadrabadi [1], Ahmadreza Asgharpourmasouleh [2] and Mark Goh [3]

1 Industrial Management Department, Yazd University, Yazd 8915818411, Iran;
Negar.Jalilian@stu.yazd.ac.ir (N.J.); alireza_naser@azd.ac.ir (A.N.S.)
2 Department of Sociology, Ferdowsi University of Mashhad, Mashhad 9177948974, Iran; asgharpour@um.ac.ir
3 NUS Business School and The Logistics Institute-Asia Pacific, National University of Singapore,
Singapore 119613, Singapore; bizgohkh@nus.edu.sg
* Correspondence: Zanjirchi@yazd.ac.ir

**Abstract:** Improving the service processes of banks requires updating and optimizing banking supply chain processes as well as understanding the interaction between the constituent elements of each process. This paper applies agent-based modeling to configure selected processes of Tejarat Bank's service supply chain, a large bank in Iran. Tejarat's banking process system is presented and the most effective processes of the banking supply chain are identified. The interactions between the processes are examined and the research model is presented. We execute the agent-based modeling in a simulated environment. The simulation results are analyzed using Taguchi's experimental method. Selected combinations of the elements of the integrated marketing communications program are presented to reduce the liquidity risk of Tejarat Bank.

**Keywords:** service supply chain; liquidity risk; agent-based modeling; Taguchi; Iran

## 1. Introduction

Industry observers note that the service sector of many countries holds a special value in employment, increasing national wealth and improving firm competitiveness in the marketplace [1]. The literature already asserts that the driver for economic growth for developed and developing countries depends on the activities of the service sector [2–4]. As such, there is a paradigm shift in the service-oriented industries [5]. Measures such as the further development of the service industries through the use of new ideas and innovations, the introduction of new services, the use of new strategies, and the redesign and proposition of new configurations can be employed to increase the productivity of services [6]. Despite the state of research on service marketing, service operations management, service science, and service engineering, there are many unresolved problems in the management of service supply chain processes as well as service delivery [5]. In short, little research has been conducted on the service supply chain and, as a result, researchers lack the necessary knowledge on the structure of the service supply chain. Regarding that the development of service-oriented industries has been an increasing trend, and since the success of the service industry depends on the optimal functioning of the supply chain, it is necessary to formulate the known behaviors in supply chain processes and to examine the dynamics of the service supply chain in different conditions [7,8].

Banks and credit institutions are service provider organizations that organize transfers of funds and expand markets through facilitating trade exchanges. Banks are the main element of the management and guidance of funds towards production units. The adjustment of cash flow is important to the economic growth of societies because, in addition to providing financial services and the capital needed, they also allocate financial resources to different production units to enhance economic growth [9]. Today, factors such as

technological advancements and international economic development have increased competition in the marketplace. In addition, entry into the competition of global monetary and banking markets has left the banking industry with numerous challenges [10]. Iranian banks are also faced with many challenges and problems, some of which are due to the macroeconomics of the country. Furthermore, some problems may confront the economic system and, consequently, the banking system. In these cases, the banking system uses its internal capabilities and employs suitable policies to resolve problems and economic crises.

Service provision in the competitive environment of the banking industry of Iran has always been affected by existing uncertainties, technological advancements, constant changes in customer needs, and, especially, political and economic events. Therefore, the development of banking activities and their success in the country require the use of modern, fast, and responsive banking operations so that by analyzing the results obtained from uncertain processes, more customers can be attracted to participating, and the competitive position of the banking system in Iran can be determined, which positively affects the profitability or the competitive position of the organization in the market. However, these will be realized if the key elements and components of the processes of the banking service supply chain, such as the information technology (IT) management process [11], customer relationship management [12,13], risk management [14,15], and so on, are identified. Moreover, by concentrating on the conditions of the system, interactions between these elements and components can be determined. Despite the importance of this fact, most studies (e.g., [16–18]), in examining organizations' performance, have focused on the banking system processes in definite environments and have neglected the environment's dynamics as well as the uncertainties of customer behavior. The reason for this is that we are dealing with unknowns and there are no solutions for them as a result. Because these studies have not considered the uncertainties of the banking systems in the real world, the results of such studies are not applicable to the strategic planning of organizations.

Therefore, given the increasing importance of the banking system in Iran and the need to generalize the processes of this system for uncertain environments, the present study used an agent-based simulation method and configured selected processes of the service supply chain in the banking system of Iran. The present research was conducted as a case study of Tejarat Bank of Iran. Tejarat Bank is one of the largest and most influential commercial banks in Iran, which benefits from rich human resources and suitable technical infrastructure. The bank provides many banking services, including e-banking and international banking services. It is one of the most important economic elements in Iran, as it organizes the receipts and payments to facilitate trade and commerce, and directs the savings and deposits towards production firms, thereby contributing to economic growth.

In recent years, new achievements in the area of banking have created a highly competitive market. The improvement in the current situation of the banking industry requires that the procedures and processes of the supply chain of the banking system be fundamentally reviewed. So far, limited studies have been conducted on the bank service supply chain, and none of the proposed models have taken into account the customer's central role as the main feature of the service, the processes involved in the bank service supply chain, and the existing interactions and inter-process communications. The aim of this study was to provide a model for the service supply chain of Tejarat Bank, which was accomplished by examining the architecture of selected processes of the banking system.

Many studies are conducted to configure, simulate, and model the supply chain and its processes. The most widely used methods in these studies are the system dynamics approach [19], discrete event simulation [20,21], statistical models [22,23], and optimization methods [24,25]. For example, Langaroodi and Amiri (2016) used the system dynamics approach and proposed a model for supply chain design with multiple levels, multiple products, and multiple regions [26]. Poles et al. (2013) explored the dynamics of the reproduction process of products in the supply chain. To this purpose, they used the system dynamics approach and simulated and modeled the inventory and production systems of the supply chain [19]. The system dynamics approach first identifies the problem

variables and then models them; it is widely used in the simulation of different problems. It provides managers with "if/then" scenarios relating to the problem [27]. However, this approach can be used only when predetermined processes of the model are considered and there is no change in the structure or relationships of the model. This approach also requires identifying processes that are sometimes hard to understand. In discrete event simulation, it is very difficult to model complex processes with multiple decision levels, and a definition of a set of constant processes and relationships at the beginning of the simulation limits the analysis of the processes and relationships that may appear during the simulation [28]. In fact, this type of simulation is simple and leads to unrealistic results [27]. The use of statistical models, especially structural equation modeling (SEM), in supply chain management has increased significantly. Kaufmann and Gaeckler (2015) reviewed 75 articles published between 2002 and 2013 on SEM and its application in the modeling of supply chains; they found that research has not fully exploited the capabilities of SEM [29]. Generally, statistical methods are applicable when the system structure is time-invariant and no model details are needed. In addition, these statistical models are unable to examine if/then scenarios [28]. Truong and Azadivar (2005) combined integer programming and a genetic algorithm to optimize a supply chain configuration design [30]. Lin et al. (2013) presented a framework to optimize the multi-objective simulation of hospital surgical services, which increased the speed of the genetic algorithm in the search spaces. This method can be used when the data and structure of the problem are known and well-defined [31]. Agent-based simulation and modeling can help to better understand the systems that consist of many interacting factors. The agent-based approach is able to examine very complex factors and their interactions and put them into computational frameworks. Its application is suggested for systems in which the factors are able to learn and gain experience, are dynamically linked to each other, and are involved in creating dynamic and strategic behavior [28–32]. Behdani et al. (2010) employed agent-based modeling and simulated the supply chain network in oil refining firms to develop and improve the supply chain performance and to mitigate disruption risks [33]. Costas et al. (2015) used agent-based modeling to investigate how the theory of constraints reduces the bullwhip effect [34]. Hence, Dev et al. (2016) integrated discrete event simulation with agent-based modeling to reconfigure the supply chain of mobile phone production [35]. Earnest and Wilkinson (2018) proposed a model based on the proportionality of products with market demand and supplier partnership [36]. In addition, Puche et al. (2019) used the Kanban method, lean production, theory of constraints, and agent-based simulation to evaluate the performance of a four-channel supply chain [37].

Unlike the methods discussed earlier, agent-based simulation and modeling are used to understand systems that consist of a large number of interacting agents. The agent-based method considers agents and the interactions between them at any level of complexity, and places them into computational frameworks. The agent-based method is used when in the system and the agents are dynamically connected to each other and participate in the formation of dynamic strategic behaviors, in addition to having the power to learn and gain experience. Despite the fact that the agent-based simulation method has superior features over other simulation and modeling methods, such as system dynamics, innovative algorithms, and so forth, and can also be used in production and operations management, only a few studies have benefited from this method. Regarding its benefits, it is, therefore, necessary to expand the application of this method to service operations management. Furthermore, the studies that have been done so far on service supply chains are limited, and their focus has been mostly on the production service supply chain; thus, there is a need for in-depth studies on the service supply chain. Therefore, it can be acknowledged that the contribution of the present study is from two perspectives. It is hoped that the results of this study can create new insights in service-oriented industries. The purpose of this study was to scrutinize the architecture or so-called configuration of selected service supply chain processes in the Iran Tejarat Bank (ITB) system by examining the selected processes of the bank service supply chain, clarifying how the elements interact in each

of the selected processes, and using the agent-based simulation method. The purpose of configuring and architecting the selected processes of the service supply chain is to achieve the functional goals of each of the selected processes in the banking system. Additionally, the purpose of the scenario writing approach is to facilitate the analysis of the results of uncertain processes, which can be helpful in determining the competitive position of the banking system. For fulfilling the aim of this study, at first, all the processes in the bank service supply chain and the set of activities performed in this industry were identified to obtain an overview of the service delivery process in the target industry. Then, the conceptual research model was developed. Next, according to the purpose of the model, the agent-based modeling cycle started, during which actions such as formulating the research question, collecting relevant hypotheses, determining the structure of the model, using and implementing the model, and the model analysis were carried out. Finally, after validating the model, we analyzed the results and presented a report on them.

## 2. Materials and Methods

The present research, in terms of purpose, variables control, and the data collection method, is a developmental, quasi-experimental, and survey-based study. From the perspective of methodology and problem-solving, the present study used the modeling and agent-based approach. Today, the agent-based modeling approach is recognized as the third scientific method after induction and analogy, because agent-based modeling is initiated with a set of basic assumptions (i.e., inductive approach) and generates simulation data for analysis (i.e., deductive approach). Hence, some experts believe that ABM, instead of being placed in the group of inductive or deductive approaches, should be placed in a new group called "productive approaches" [38].

The population of the present research included all of the active managers, deputies, and experts of managerial units of Tejarat Bank of Iran. Since the research methodology required interviews and consulting sessions with the banking experts, members of the consultation group were sampled using judgmental sampling from experienced members and experts of the banking sector with at least 5 years of work experience and a bachelor's degree or higher. For this purpose, we first visited the ITB studies and planning office located in Tehran province and the supply deputy in Yazd province, and the deputies were asked to introduce qualified people as members of the consultation group. A total of 13 members were selected from Tehran's management units and 8 members were selected from Yazd's management units. To fulfill the research purpose, the following research questions were formulated.

(1) What are the main processes by which services are produced and used in the bank?
(2) What processes are recognized as selected processes of the service supply chain in banking?
(3) What is the relationship between the selected processes of the banking service supply chain work?
(4) How can we explain the operational model based on the existing communication and interaction between the banking service supply chain processes?

Figure 1 shows the research steps.

The processing system was designed in three steps. At first, the supply chain processes for ITB were identified. We referred to the management units of ITB, reviewed the job descriptions and upstream documents of the banking units, and interviewed the active people in the management units (at the levels of manager, deputy, and expert) who were also members of the panel. Second, the process framework of ITB was set, based on the PCF process framework. The extracted processes in the first step were then set in the framework of the PCF model for banking processes, so the extracted processes and PCF model processes that were not mentioned in the panel discussions, along with the sub-processes, were formed into a questionnaire. Using the Delphi method, we discussed it again with the panel of bank experts for review. After the panel, the items that were irrelevant and disproportionate to ITB's processes were removed, and the necessary items

were added to the questionnaire so that the process framework for ITB could be designed and compiled based on the PCF process framework. Lastly, the final framework was introduced for ITB's service supply chain processes.

**Step One: Design Process System of Banking service supply Chain**
- Identification of bank's service supply chain processes
- Adjusting the Bank's Process Framework based on the PCF Process Framework
- Finalizing the process system of the bank's service supply chain

**Step two: Determine selected processes in banking service supply chain**
- Design DEMATEL questionnaire and collecting comments
- Convert collected comments to trapezoid intuitive fuzzy numbers
- Data analysis
- Present results of intuitive fuzzy DEMATEL analysis to experts for confirmation

**Step three: Explain research conceptual model**
- Identify relationships between selected processes of banking service supply chain
- Identify conceptual categories and major factors affecting customer behavior
- Designing and explaining the conceptual model of the research

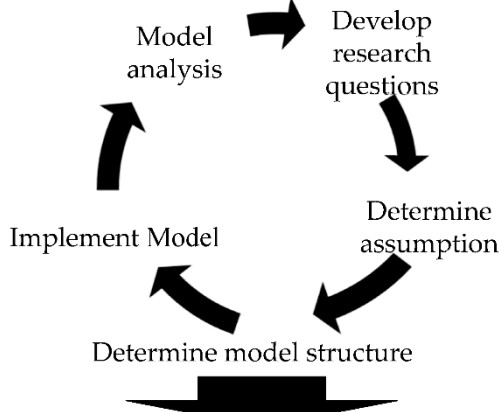

**Step 5: Analyzing the results and presenting report**
- Developing scenarios
- Implementing scenarios in simulation environment
- Results analysis and reporting

**Figure 1.** Research process.

The second stage was performed in five steps. Based on the framework of processes designed in the previous stage, a questionnaire was developed to identify the relationships within the service supply chain processes in ITB. The questionnaire was then distributed among the respondents. After the data were collected through the questionnaire, the respondents' opinions were transformed into trapezoid intuitive fuzzy linguistic variables. In the next step, the collected data were analyzed by the intuitionistic fuzzy DEMATEL methodology, and the most effective processes of the Tejarat Bank service supply chain were introduced. Lastly, the results of the intuitionistic fuzzy DEMATEL analysis were presented to experts for confirmation and finalization.

The third stage was the development of a conceptual research model. First, in order to identify the structure and purpose of each of the selected processes of the Tejarat Bank service supply chain, the members of the panel were interviewed, and articles published in reputable scientific databases were read. Content analysis was employed to determine the most important factors affecting the selected processes of ITB. In order to extract the most important effective factors, the articles related to the subject were read, and the members of the panel were interviewed. After extracting and coding the factors, we decided on suitable themes for the conceptual categories. The result for this stage was a set of the most important factors affecting the selected processes of the Tejarat Bank service supply chain. According to the structure and purpose of the selected processes and the result of the content analysis, a conceptual research model was designed. This model was presented to the members of the panel of experts to receive their opinions in order to finalize the conceptual research model and to be used in the fourth step.

Stage four, designing the agent-based modeling cycle, included five steps: (a) formulating the research question, (b) compiling the research hypotheses, (3) determining the structure of the model, (4) applying and implementing the model, and (5) analyzing the model. The modeling began with a clear and precise research question about the conceptual model. In the second step, the related hypotheses were formulated for the processes and structures involved in the conceptual research model. Then, the factors involved in the selected supply chain processes were identified and the behavior of each factor and the modeling environment were defined. Some behaviors of the factors involved in the model were explained by the design approach of Taguchi experiments. In the application and implementation step of the model, an attempt was made to define the factors involved in the model in the form of moving objects and the behavior of the factors in the form of mathematical relations and to program them into the Anylogic 8.8.2 software.

Stage five involved analyzing the results and reporting the findings. At this stage, the results of the implementation of the scenarios in the fourth stage were analyzed and suggestions were made in order to improve the bank service supply chain processes and to develop the model.

## 3. Results

### 3.1. Design Process System of Banking Service Supply Chain Subsection

The bank's job description and upstream documents were examined and members of the consultation group were interviewed to establish the bank's processes. From the interviews, processes of marketing and sales management, risk management, branch and state administration, human capital management, IT management, financial management, and business improvement management were identified as the main processes. Thus, their sub-processes were extracted. Next, the extracted processes were adjusted within the framework of the process classification model so that the extracted processes and processes of the PCF (Process Classification Framework) model, which were not mentioned in the interviews, along with sub-processes were listed and then were revised in consultation with the experts. After three consultations, irrelevant and inappropriate items were removed and needed items were added to the list. Finally, the final system of the service supply chain processes of the banking system at ITB was formed. The processes of the ITB supply chain were divided into two categories: (1) operational processes (e.g., planning

and formulation of strategies, designing services and products, marketing and the sales of products and services, product delivery, service delivery, and public relations and advertising management), and (2) support and management processes (e.g., financial resources management, human resources management, IT management, risk and credit management, the bank's branch and state administration, marketing and sales management, and business improvement management). Figure 2 shows the process system of the bank's service supply chain, arising from the interviews and consultation with the experts.

| Operational processes | | | | | |
|---|---|---|---|---|---|
| Planning and formulation of strategies (1) | Designing services and products (2) | Marketing and the sales of products and services (3) | Product delivery (4) | Service delivery (5) | Public relations and advertising management (6) |
| Support and management processes | | | | | |
| Financial resources management (7) | | | | | |
| Human resources management (8) | | | | | |
| IT management (9) | | | | | |
| Risk and credit management (10) | | | | | |
| The bank's branch and state administration (11) | | | | | |
| Marketing and sales management (12) | | | | | |
| Business improvement management (13) | | | | | |

**Figure 2.** Process framework of Tejarat Bank's service supply chain.

### 3.2. Determine Selected Processes of Service Supply Chain

The selected process in the present study is a process that has the greatest impact on other organizational processes. In order to determine the most effective processes of ITB's service supply chain, we followed the steps of implementing the DEMATEL method [39] and considered intuitive fuzzy relationships [40] to design a matrix composed of the ITB service supply chain processes. The respondents were asked to determine the type and extent of their relationship to other processes with respect to each process and their pairwise comparison. At this stage, we consulted with 12 members of the consultation group, which were selected by judgmental sampling. The questionnaires were completed and collected and the respondents' views were converted into intuitive trapezoid fuzzy linguistic variables and, accordingly, a matrix of intuitive primary fuzzy direct relations was created. Afterwards, the arithmetic mean of the experts' opinions was calculated, the mean of the intuitive fuzzy direct relationships was obtained, and the matrix of normalized intuitive fuzzy direct relations formed. Next, the identity matrix ($I_{13*13}$) was formed and all the values relating to it were subtracted from the corresponding values in the matrix of normalized intuitive fuzzy direct relations. Then, the invertible matrix was obtained by subtracting the identity matrix from the matrix of direct intuitive fuzzy normal relations. By multiplying the normal matrix by the invertible matrix, the total relation intuitive fuzzy matrix was obtained. Next, values existing in the total relation intuitive fuzzy matrix were determined and the final total relation matrix was obtained. Finally, given the direct and indirect relationships governing the processes, these processes were rated based on their degree of impact and the extent to which they interact with other processes. *D* represents the total impact on the process and *R* indicates the total impact on each of the supply chain

processes. To calculate the impact of each process and its interactions with other processes of the bank service supply chain, the values of D − R and D + R were calculated.

Values of (D − R) and (D + R) were sorted in descending order and, accordingly, the most effective and the most interactive processes of ITB's service supply chain were extracted. According to the obtained results, the processes of goods delivery, risk management, IT management, public relations and advertising management, and marketing and sales management had the greatest impact on the processes of the banking service supply chain. In addition, the processes of business improvement management, marketing and sales management, branch and state administration, product marketing and sales, and human capital management had the greatest interactions with other processes of the supply chain.

*3.3. Research Model*

The results of our study were compared with previous studies [41–43], which have emphasized the effect of processes of IT management, risk management, and public relations and advertising management on other banking processes. Some other studies (e.g., [44,45]) have emphasized the effect of the risk management process on other banking processes. Yusheng and Ibrahim (2019) showed a significant impact of the product delivery management processes and service delivery management processes [46]. Theodosiou et al. (2012) confirmed a significant impact of the processes of marketing and sales management and public relations and advertising management [47].

The selected processes were discussed in the consultation meetings and, according to the PCF framework of Tejarat Bank, the problems of the banking space, and comments of the members, the effective processes were confirmed and the conceptual model of the research was explained. Within the PCF framework of Tejarat Bank, the process of product delivery management is focused on the products that are designed and produced for the purpose of advertisement and providing information about ITB services. The consultation group members expressed that the mentioned process was significantly related to the marketing and sales process and the public relations and advertising process. It was also stated that banking services such as ATMs, mobile banks, and telephone banks, which are presented in the form of ITB products alongside its services, need an IT platform and hence, they affect the current activities of the IT management process. The consultation group members examined the IT management process within the PCF framework and stated that the type of organizational activities, information volume, and the banking products is such that a wide range of organizational activities flows in the context of IT, and hence, the quality of the IT process management affects the quality of other banking processes. Regarding the impact of the banking risk management process, it was stated that Tejarat Bank of Iran has always attempted to formulate strategies and programs to improve performance indicators such as resource allocation (deposits), facility granting, level of receivables, providing new e-banking services, letters of credit (liabilities) and earnings (losses), and so on, to realize them as well as possible. The liquidity risk is one of the most common risks that the ITB system faces, which, because of its connection with the credit risk, is of particular importance. The emphasis on liquidity risk management has led to deposits (absorbing resources) being seen as the most important performance indicator and to solutions being proposed to increase the bank's cash flow. Encouraging customers to deposit in the bank is a solution to absorb banking resources. Among the selected processes, measures related to the processes of public relations and advertising and sales and marketing management can increase bank liquidity, and thus, improve the bank's performance on absorbing resources and reducing liquidity risk. As ITB is keen to reduce liquidity risk, the research model was formulated with the proposed inter-process relationships and the incorporation of the processes of risk management, public relations and advertising, and marketing and sales management. This model is based on the IT and product delivery management process being integrated with the process of public relations and advertising management.

The present study, with respect to the above-mentioned conceptual model, employed an integrated marketing communication (IMC) program, the formulation of which, in turn, is a sub-process of the marketing and sales management process, and with a keen understanding of the characteristics of the customers in the target markets, increases the deposit rates by target markets. The research model, after applying the changes proposed by the consultation group, was confirmed. Figure 3 presents the research model.

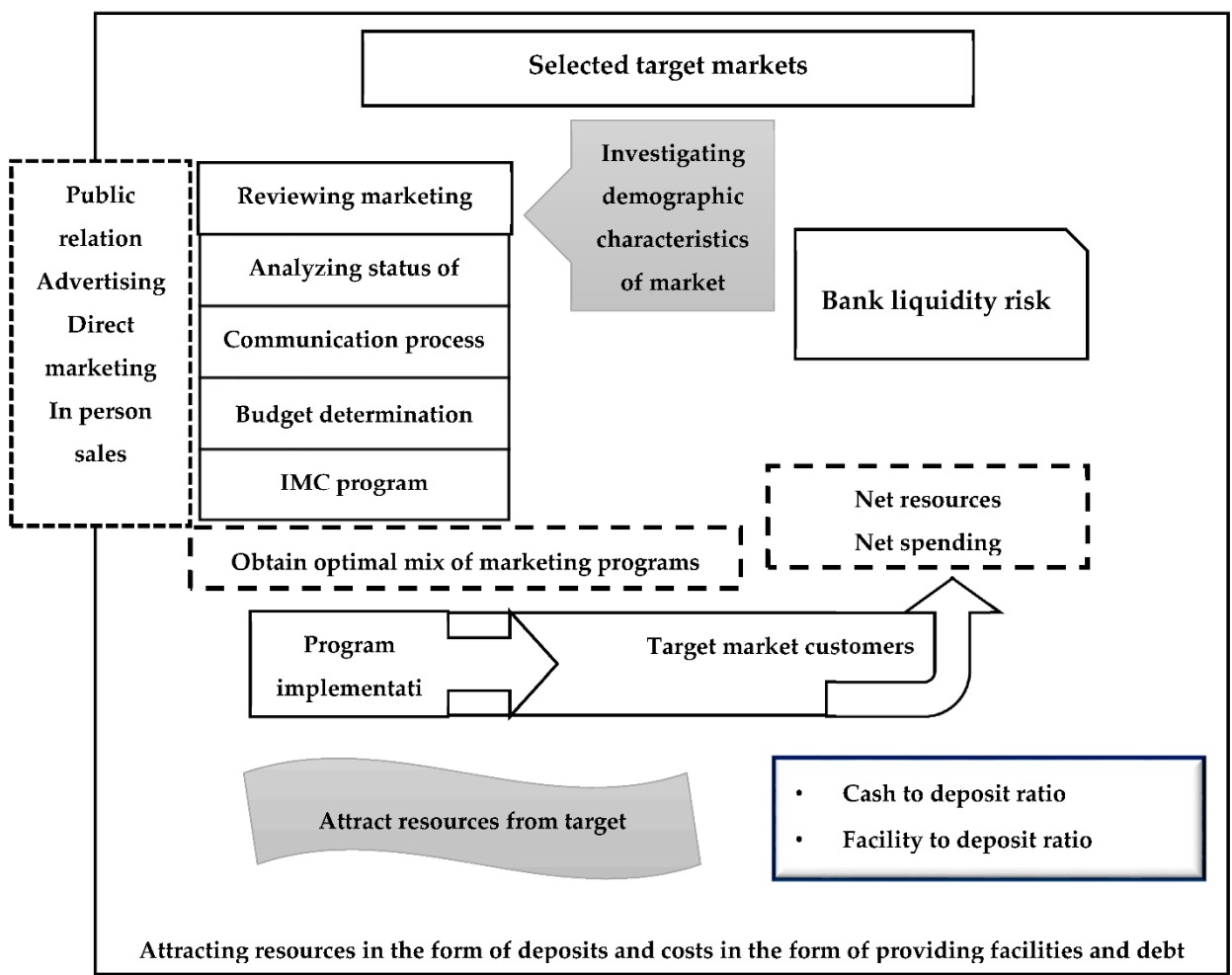

**Figure 3.** Research model.

Regarding the conceptual model of the present research, the service supply chain processes of Tejarat Bank were identified and extracted through the comprehensive classification model for processes and the expert opinion in Tejarat Bank to comply with the process dimension of the service supply chain, which is emphasized in all models. The selected processes of the Tejarat Bank service supply chain were identified through a multi-criteria decision-making method, based on inter-process relationships and interactions. For the confirmation of the service supply chain model proposed by Sang and Yu (2009) and Lin et al. (2010), which emphasizes the existence of a network and communication structure in the service supply chain, the existing relationships and interactions among the selected processes were explained by the expert opinions of the members of the panel, and the structural and communication dimension of Tejarat Bank's service supply chain was obtained. Because the central role of the customer as the main characteristic of the service is not emphasized in any of the proposed models, we devised a set of conduct rules for customers and included them in the model to further emphasize the role of the customer in the management of the supply chain of Tejarat Bank. The study of service supply chain

models discussed earlier shows that the dynamic nature of service and the processes in the service supply chain had not been cared for. Thus, in the present study, we implemented a conceptual model and agent-based modeling and simulation to take care of the dynamic nature of the Tejarat Bank service supply chain.

To identify the antecedents of customer decisions to deposit and receive loans and facilities, content analysis was used. A database containing a collection of relevant articles published in reputable scientific databases was created. To strengthen the database, some interviews were conducted with the consultation group members; they were asked to state their views on the factors affecting customer behavior on depositing and lending. The database included 60 articles and 12 transcripts obtained from the interviews. Then, expressions indicating the influential factors on customer behavior through careful study of the database's texts, and based on the article numbers, interviews, and intended factor class (i.e., factors affecting deposits or receiving loans and facilities) were coded and then were analyzed in the form of conceptual categories. Finally, examining the conceptual categories led to the extraction of relevant themes. Regarding our purpose of identifying the individual factors affecting the bank customers' decisions, the themes extracted from the opinions of the consultation group were categorized as the following: (1) environmental factors, and (2) individual factors. Table 1 contains the results of the content analysis.

### 3.4. Agent-Based Modeling

The present study aimed to develop a model to analyze the effect of the IMC program on liquidity risk. The first phase of the agent-based modeling cycle examines the following research question.

- Does implementation of the IMC program affect ITB's liquidity risk?

Some interviews were conducted with the consultation group members to identify the ITB academic target market. During the interviews, it was revealed that universities are one of the most important ITB target markets. Universities have three populations, including students, employees, and faculty members, and compared to other target markets of the bank, it has great diversity in terms of demographic and behavioral characteristics. We referred to the research and planning institute of the Ministry of Science, Research and Technology of Iran, the main authority of statistics of Iran universities, to extract the number of academic customers of Tejarat bank. It was revealed that out of 129 universities working with Tejarat Bank, the population of academic customers including students with associate degrees, BSc, MSc, and Ph.D. degrees is 503,201. In addition, the population of all employees is 27,008, and the population of professors, including instructors, assistant professors, and associate professors, is 15,035. It was also discussed that the IMC program includes a set of sales promotion tools, including advertising, direct marketing, digital marketing, in-person sales, sales progress, and public relations that are included in the university's IMC program based on the demographic and behavioral characteristics of students, employees, and faculty members. Therefore, the bank was considered the modeling environment, and the academic clients were considered interactive agents who operate in the bank's environment and interact with their environment through the tools of the IMC program. Each of the sales promotion tools has a certain effectiveness rate, which shows to which extent their implementation encourages customers to use the bank's products. The bank's customers communicate with each other at a specific call rate and anyone can encourage the individuals connected with them to use ITB services.

The present study implemented the conceptual model of the research in AnyLogic 8.8.2. To this purpose, it considered Tejarat Bank as the modeling environment and divided the bank's customers as the model agents into three groups: students, professors, and employees. Since agent-based modeling factors interact with the environment and other factors, it was necessary to specify how each factor behaves in each population. Hence, the behavioral hierarchy is a state diagram for each population. Each factor population participating in the model is affected by a set of promotion tools in the IMC program with a given effectiveness rate within a defined time period, and after absorbing the effect of

the promotion tools each factor can convert to a potential agent for deposits and receiving banking facilities. Figures 4–6 display the state diagrams for the factor population of students, employees, and professors.

**Table 1.** Results of content analysis.

| | Theme | Conceptual Categories Affecting Customer Decisions Based on Deposits |
|---|---|---|
| **Individual factors** | Customer age | Bank customer age |
| | Customer income | Customer income level |
| | Education | Customer education level |
| | Account history | History of customer's use of the bank's services |
| | Customer experience | Customer's experience in receiving services |
| | Customer satisfaction | Customer's satisfaction with the bank's services |
| **Environmental factors** | Facilities | Amount of paid credits, diversifying the deposit headings |
| | Branch structure | Enough space in branches, the existence of furniture in the waiting room, facilitating the car park, presence of ATMS, facilitating access to account status, facilitating access to the essential items, availability of appropriate physical facilities |
| | Branch location | Quick access to customers, beliefs of region's residents, customers' location, competitors' conditions, cultural environment governing the region, religious perceptions of the region's residents, presence of the bank's branches in Iran |
| | Branch beauty | Branch's appearance indicators, including psychological, physical, and social aspects, branch decorations, branch space aesthetics |
| | Service diversity | Improving and diversifying services, managing relationships with customers |
| | Service quality | Provide quality banking services, meet customer expectations by providing quality products, deliver desirable services |
| | E-banking | The use of new technologies, e-banking services, e-banking training |
| | Deposit interest | Over recovery of long-term investment deposits, deposits on account earnings, deposits' defined earnings |
| | Rewards | Lottery, rewards intended for the customer account |
| | Speed and accuracy | Speed and accuracy of employees in providing services |
| | Staff behavior | Employees' individual capabilities in providing services, employees' communication skills, the way employees deal with customers |
| | Population index | GDP, inflation rate, government monetary and financial policies, political stability, laws, housing price, currency rate, gold coin price, car price, national income, financial markets boom |
| **Individual factors** | Customer age | Customer age |
| | Customer income | Customer income, recipient occupation, activity type of the loan recipient |
| | Education | Customer's education level |
| | Account history | Customer's history of service use |
| | Customer experience | Customer's experience in receiving the banking services |
| | Bounced check | Number and amount of bounced checks before receiving loans |
| | Collateral | Customer ownership status, current status of customer's assets |
| **Environmental factors** | Repayment schedule | Loan repayment period |
| | Number of installments | Number of payments |
| | Collateral type | Type of collateral required |
| | Installments amount | Loan installment amount, loan commission rate |

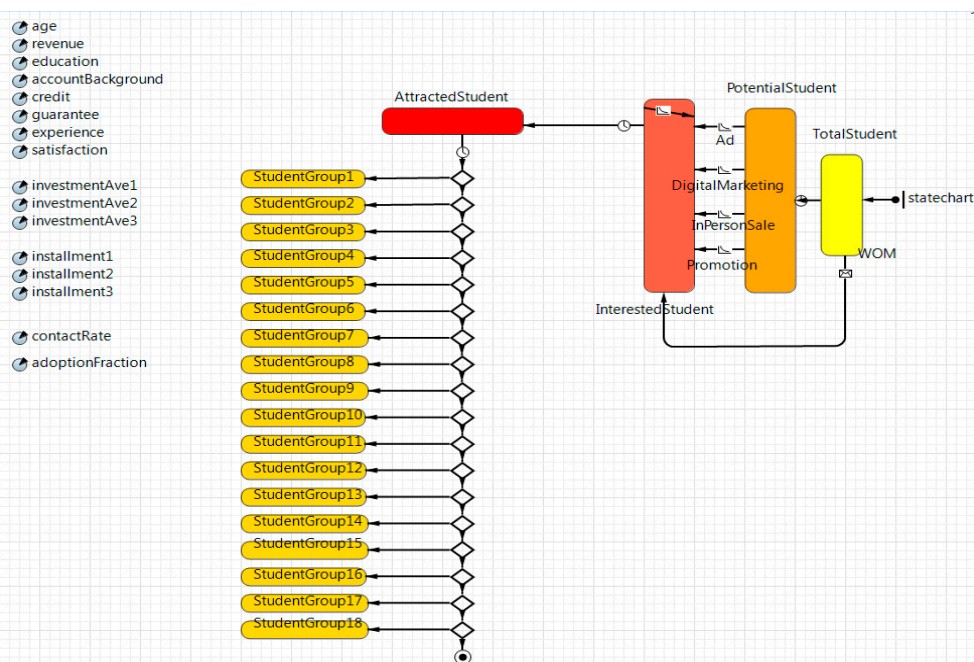

**Figure 4.** State diagram of student population.

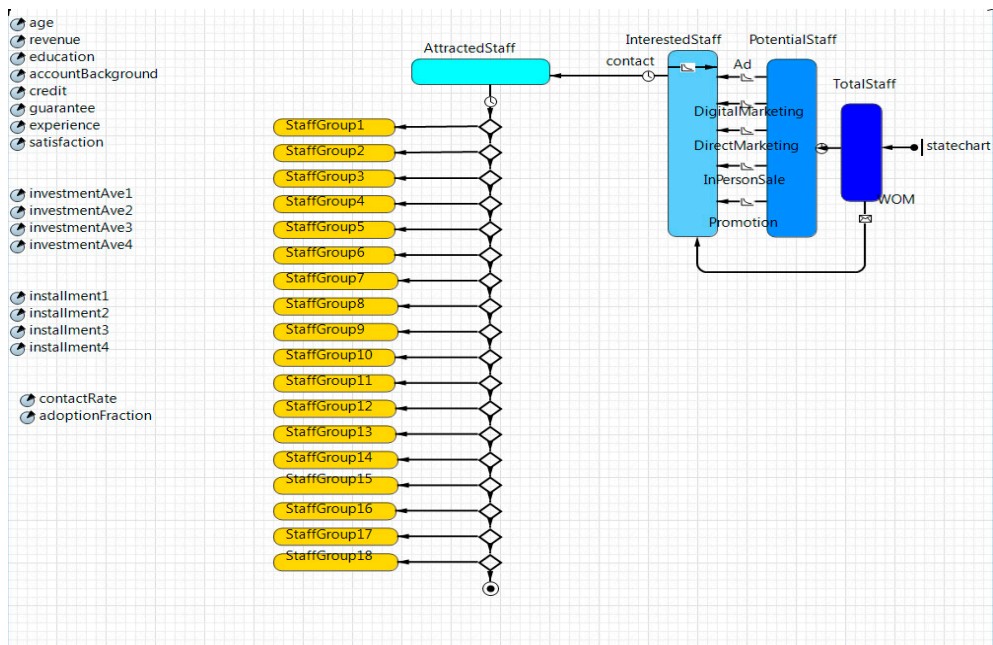

**Figure 5.** State diagram of employee population.

Next, a set of behavioral rules for the interactions of each factor was defined. Behavioral rules were defined to determine if a customer, with respect to the predefined parameters, has the necessary conditions for investment and receiving facilities. If they do and they are qualified to receive facilities, how much have they deposited and how many facilities will they demand of the bank? We considered the factors affecting customer decisions on depositing and receiving banking facilities and explain the behavioral rules with respect to the Taguchi experimental design. Factors such as age, income, education level, account history, experience, satisfaction, and ownership status affected the bank customers' decisions. Intervals for each population and each of the above factors were set based on the opinion of the panel members. Since six 3-level factors L18 (36) affect the customers' decisions on depositing and receiving banking facilities, we used the experimental design

table and extracted 18 experiments for each factor population. Then, the opinions of the panel members in each of the defined situations were collected to determine the average deposit of the customer as well as the amount of facilities requested.

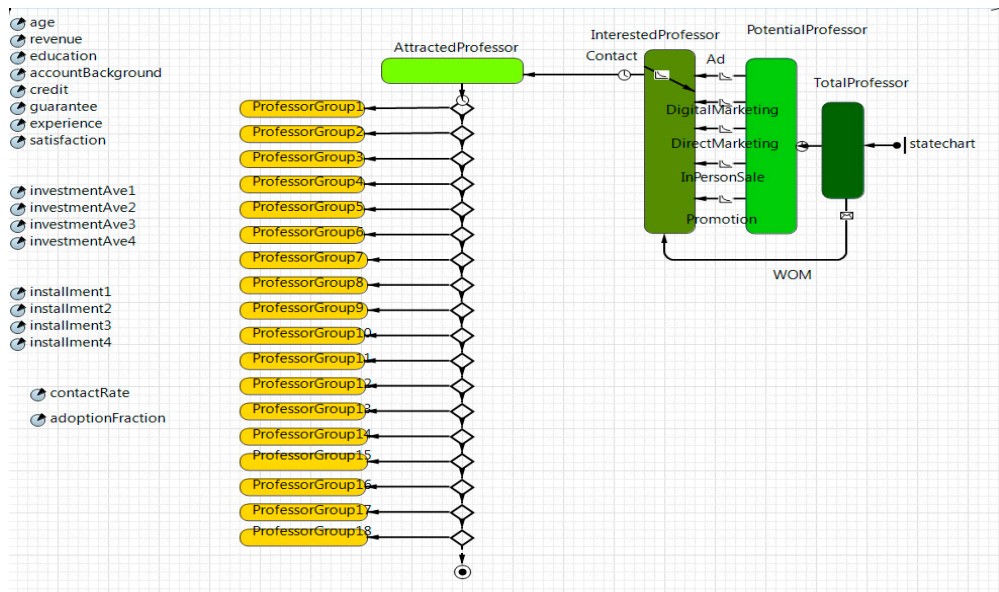

**Figure 6.** State diagram of professor population.

After describing the behavioral rules, the marketing rules were formed. The panel members felt that the rules should be formed based on the effectiveness level of the marketing tools. Four effectiveness levels for each tool were set: first level: 0–0.25; second level: 0.25–0.5; second level: 0.5–0.75; and fourth level: 0.75–1. The Taguchi experimental design method was used to extract the rules, as shown in Table 2.

**Table 2.** Marketing rules based on the Taguchi experimental design.

| Rule | Advertisement | Direct Marketing | In Person Sales | Digital Marketing | Sales Progress |
|------|---------------|------------------|-----------------|-------------------|----------------|
| 1 | 0–0.25 | 0–0.25 | 0–0.25 | 0–0.25 | 0–0.25 |
| 2 | 0–0.25 | 0.5–0.25 | 0.5–0.25 | 0.5–0.25 | 0.5–0.25 |
| 3 | 0–0.25 | 0.75–0.5 | 0.75–0.5 | 0.75–0.5 | 0.75–0.5 |
| 4 | 0–0.25 | 1–0.75 | 1–0.75 | 1–0.75 | 1–0.75 |
| 5 | 0.5–0.25 | 0–0.25 | 0.5–0.25 | 0.75–0.5 | 1–0.75 |
| 6 | 0.5–0.25 | 0.5–0.25 | 0–0.25 | 1–0.75 | 0.75–0.5 |
| 7 | 0.5–0.25 | 0.75–0.5 | 1–0.75 | 0–0.25 | 0.5–0.25 |
| 8 | 0.5–0.25 | 1–0.75 | 0.75–0.5 | 0.5–0.25 | 0–0.25 |
| 9 | 0.75–0.5 | 0–0.25 | 0.75–0.5 | 1–0.75 | 0.5–0.25 |
| 10 | 0.75–0.5 | 0.5–0.25 | 1–0.75 | 0.5–0.25 | 0–0.25 |
| 11 | 0.75–0.5 | 0.75–0.5 | 0–0.25 | 0.5–0.25 | 1–0.75 |
| 12 | 0.75–0.5 | 1–0.75 | 0.5–0.25 | 0–0.25 | 0.75–0.5 |
| 13 | 1–0.75 | 0–0.25 | 1–0.75 | 0.5–0.25 | 0.75–0.5 |
| 14 | 1–0.75 | 0.5–0.25 | 0.75–0.5 | 0–0.25 | 1–0.75 |
| 15 | 1–0.75 | 0.75–0.5 | 0.5–0.25 | 1–0.75 | 0–0.25 |
| 16 | 1–0.75 | 1–0.75 | 0–0.25 | 0.75–0.5 | 0.5–0.25 |

We created several variables in the final phase of building the model: the amount of deposits, amount of received facilities, and cash balance of the bank. The model was run in a one-year time window. All the 16 marketing rules were implemented into the simulation environment 10 times, along with the set population, with respect to their behavioral rules and the effectiveness of each of the IMC tools to deposit and receive loans and banking facilities. The amount of deposits (bank resources) and amount of loans and facilities given to customers (bank consumption) were saved at the end of the simulation run. From the above values, and considering the cash balance of ITB at the end of 2017, the bank's liquidity risk, with respect to two liquidity risk indicators, was found.

*3.5. Data Analysis*

The 16 marketing experiments for each of the two liquidity risk indicators were conducted separately in the agent-based simulation environment, and the values obtained from the Taguchi experimental method as well as the outputs of the Qualitek-4 software were analyzed. Then, the ratios of utility values to poor performance were calculated with respect to the calculated liquidity risk values for each of the marketing rules; the results are shown in Table 3. According to the shape of the loss functions and to create the best conditions, the largest numerical value of the signal-to-noise ratio is ideal.

**Table 3.** Values of the S/N ratio based on the liquidity risk index (bank's cash to total deposit).

| Marketing Rule | S/N | Marketing Rule | S/N |
|---|---|---|---|
| 1 | −137.722 | 9 | −137.909 |
| 2 | −139.229 | 10 | −137.818 |
| 3 | −163.626 | 11 | −137.72 |
| 4 | −137.693 | 12 | −137.827 |
| 5 | −137.69 | 13 | −137.967 |
| 6 | −137.725 | 14 | −137.789 |
| 7 | −137.726 | 15 | −137.715 |
| 8 | −137.681 | 16 | −137.931 |

**Values of the S/N ratio based on the liquidity risk index (received loans and facilities to total deposit)**

| Marketing Rule | S/N | Marketing Rule | S/N |
|---|---|---|---|
| 1 | −22.791 | 9 | −23.594 |
| 2 | −31.707 | 10 | −21.228 |
| 3 | −52.856 | 11 | −20.173 |
| 4 | −23.37 | 12 | −23.603 |
| 5 | −21.36 | 13 | −29.459 |
| 6 | −19.733 | 14 | −21.894 |
| 7 | −22.613 | 15 | −23.087 |
| 8 | −19.894 | 16 | −21.17 |

### 3.5.1. Determine Main Effects and Appropriate Levels of Advertising Tools in IMC Program

The Taguchi method was used to analyze the liquidity risk and to find the corresponding values of the main effects. The main effect of factor A at level L is equal to the sum of responses at that level divided by the number of responses. Table 4 shows the main effect of the IMC program of ITB at different levels.

From Table 4, |Lmax − Lmin| for advertising exceeds the other tools, that is, this factor is effective in encouraging customers to get loans and facilities and attracting depositors, thereby reducing the liquidity risk of the bank. Further, as the response type is "the lower the liquidity risk, the better", the appropriate level of each tool of an IMC program

corresponds to a lower absolute value for the main effect. Table 5 shows the levels for each of the IMC program tools with respect to the values of the main effects.

**Table 4.** Main effect of IMC tools with respect to the first liquidity risk index.

| Marketing Tool | Level 1 | Level 2 | Level 3 | Level 4 | LMax-LMin |
|---|---|---|---|---|---|
| Advertisement | −144.598 | −137.708 | −137.817 | −137.851 | 6.89 |
| Direct marketing | −137.835 | −138.157 | −144.199 | −137.783 | 6.416 |
| In-person sales | −137.878 | −138.133 | −144.251 | −137.803 | 6.646 |
| Digital marketing | −137.781 | −138.167 | −144.265 | −137.761 | 6.504 |
| Sales progress | −137.746 | −138.219 | −144.286 | −137.723 | 6.563 |
| **Main effect of IMC tools with respect to second liquidity risk index** | | | | | |
| Marketing Tool | Level 1 | Level 2 | Level 3 | Level 4 | LMax-LMin |
| Advertisement | −32.681 | −20.9 | −22.149 | −23.902 | 11.781 |
| Direct marketing | −24.301 | −23.64 | −29.682 | −22.009 | 7.673 |
| In-person sales | −20.967 | −24.939 | −29.559 | −24.167 | 8.592 |
| Digital marketing | −22.725 | −25.308 | −29.153 | −22.446 | 6.707 |
| Sales progress | −21.75 | −24.771 | −31.413 | −21.699 | 9.714 |

**Table 5.** Selected combinations of tools of the IMC program. Level for each of the IMC tools with respect to first liquidity risk index.

| Marketing Tool | Advertisement | Direct Marketing | In-Person Sales | Digital Marketing | Sales Progress |
|---|---|---|---|---|---|
| Suitable effectiveness level | 2 | 4 | 1 | 4 | 4 |
| **Level for each of the IMC tools with respect to second liquidity risk index** | | | | | |
| Marketing tool | Advertisement | Direct Marketing | In-Person Sales | Digital Marketing | Sales Progress |
| Suitable effectiveness level | 2 | 4 | 1 | 4 | 4 |

### 3.5.2. Confirmatory Test

Confirmatory testing was performed to verify the results. Optimal levels of the IMC program tools do not exist among the 16 test modes (see Table 3). To confirm the accuracy of the software output, the experiments at the mentioned levels were repeated 10 times in the agent-based simulation environment, as shown in Table 6.

From Table 7, the S/N ratio at the optimal state for both indicators of the liquidity risk was greater than those in all the 16 previous experiments, which confirms the right choice of optimal levels of marketing tools in the IMC program based on the Taguchi method.

**Table 6.** Results of confirmatory test.

| Marketing | Direct Marketing | In-Person Sales | Digital Marketing | Sales Progress | Repetitions | | | | | | | | | | S/N Ratio |
|---|---|---|---|---|---|---|---|---|---|---|---|---|---|---|---|
| | | | | | First | Second | Third | Fourth | Fifth | Sixth | Seventh | Eighth | Ninth | Tenth | |
| 2 | 4 | 1 | 4 | 4 | | | | | | | | | | | |
| Liquidity risk (first index) | | | | | 2,091,800.805 | 2,091,800.805 | 2,091,800.805 | 925,392.5089 | 7,460,699.773 | 925,392.5089 | 925,392.5089 | 2,018,583.868 | 2,018,583.868 | 2,018,583.868 | −129.221 |
| Liquidity risk (second index) | | | | | 135.298 | 51.492 | 580.475 | 135.298 | 580.475 | 94.708 | 46.607 | 580.475 | 94.708 | 135.298 | −5.036 |

**Table 7.** Most desired combination of effectiveness level of each element of the IMC program.

| Factor | Marketing | Direct Marketing | In-Person Sales | Digital Marketing | Sales Progress |
|---|---|---|---|---|---|
| Scenario 1: 30% reduction of the deposit volume and 50% increase of loans and facilities | | | | | |
| Suitable effectiveness level | 1–0.75 | 0.5–0.25 | 0–0.25 | 0.5–0.25 | 0–0.25 |
| Scenario 2: 30% reduction of the deposit volume and 150% increase of loans and facilities | | | | | |
| Suitable effectiveness level | 1–0.75 | 0–0.25 | 0.5–0.25 | 0–0.25 | 0–0.25 |
| Scenario 3: 30% reduction of the deposit volume and 100% increase of loans and facilities | | | | | |
| Suitable effectiveness level | 1–0.75 | 0.5–0.25 | 0–0.25 | 0.5–0.25 | 0–0.25 |
| Scenario 4: 40% reduction of the deposit volume and 150% increase of loans and facilities | | | | | |
| Suitable effectiveness level | 1–0.75 | 0.5–0.25 | 0–0.25 | 0.5–0.25 | 0–0.25 |
| Scenario 5: 40% reduction of the deposit volume and 50% increase of loans and facilities | | | | | |
| Suitable effectiveness level | 1–0.75 | 0–0.25 | 0.5–0.25 | 0.5–0.25 | 0–0.25 |
| Scenario 6: 40% reduction of the deposit volume and 100% increase of loans and facilities | | | | | |
| Suitable effectiveness level | 1–0.75 | 0.5–0.25 | 0.5–0.25 | 0.5–0.25 | 0–0.25 |
| Scenario 7: 50% reduction of the deposit volume and 150% increase of loans and facilities | | | | | |
| Suitable effectiveness level | 1–0.75 | 0–0.25 | 0.5–0.25 | 0.5–0.25 | 0–0.25 |
| Scenario 8: 50% reduction of the deposit volume and 50% increase of loans and facilities | | | | | |
| Suitable effectiveness level | 1–0.75 | 0–0.25 | 0–0.25 | 0–0.25 | 0–0.25 |
| Scenario 9: 50% reduction of the deposit volume and 100% increase of loans and facilities | | | | | |
| Suitable effectiveness level | 1–0.75 | 0–0.25 | 0.75–0.5 | 0.5–0.25 | 0–0.25 |

*3.6. Scenario Analysis*

Iran's economy suggests that the rate of economic indicators over the past two to three years has significantly declined, to the point that there is a significant gap between the direct investment indexes in Iran and the rival countries. In addition, the rate of fixed capital formation in Iran is becoming negative. As a result, the continuity of the current conditions in the long run will jeopardize the dynamics and stability of the Iranian economy. In the final step, given the economic conditions of Iran and given the importance of the banking customer behavior on the amount of the liquidity risk, scenarios relating to the future behavior of customers in depositing and receiving loans and the banking facilities, with an emphasis on the university target market, were formulated and analyzed. To this purpose, some interviews were conducted with the panel members. According to the interviews, customers' abilities to deposit—as well as his or her willingness to do so—are important factors that, as the main driver, encourage customers to deposit. The current economic conditions of Iran have caused the student population to lack the ability and the professor population to lack the willingness to deposit. Similarly, ability and willingness to deposit have also decreased in the employee population, and in the near future, the amount of deposits in the university target market will decrease, causing this reduction in the bank's resources to become equivalent to the inflation rate. Therefore, considering the inflation rates of 30%, 40%, and 50%, it can be expected that in the future, the volume of deposit-making in the ITB academic target market, with a reduced rate equal to the current inflation rate, will reach up to 70%, 60%, and 50% of the current deposit rate. On the other hand, the demand for loans and banking facilities will grow significantly because all three academic populations, more than before, welcome the banking facilities, which in turn results in 50%, 100%, and 150% increases in banking spending. In order to examine the future status of the behavior of ITB academic customers, nine scenarios were formulated based on the interview results. In order to determine the most desirable combination of elements of the IMC program of ITB, the agent-based model settings, based on the values related to the reductions in the amounts of deposit-making and, at the same time, the values of the growths of the loans and banking facilities, were changed by customers in each scenario. In addition, the space depicted in each scenario, with respect to each of the marketing rules, was implemented in the simulation environment, and values relating to the liquidity risk, with respect to the second indicator of calculating the liquidity risk, were registered. The budget and time constraints prevent banks to simultaneously increase the

effectiveness levels of all elements of the IMC program to the highest level. Therefore, it was necessary to present the most desirable combination of the effectiveness level of each of the elements for each scenario. Accordingly, the results of implementing each scenario in the simulation environment were analyzed separately by an experimental design method to determine the most desirable combination of the marketing tools in the academic target market. The results are presented in Table 7.

$$S/N = -10 \ LOG \ (\llbracket 2091800/805 \rrbracket \text{^}2 + \llbracket 2091800/805 \rrbracket \text{^}2 + \cdots$$
$$+ \llbracket 2018583/868 \rrbracket \text{^}2)/10 = -129/221$$
$$S/N = -10 \ LOG \ (\llbracket 135/298 \rrbracket \text{^}2 + \llbracket 51/492 \rrbracket \text{^}2 + \cdots + \llbracket 135/298 \rrbracket \text{^}2)/10$$
$$= -5/036$$

## 4. Conclusions

The present study, based on selected processes of the banking supply chain and by using the agent-based simulation method, configured selected processes of the service supply chain in the banking system of Iran. To this purpose, we followed the PCF process model and identified all the processes and sub-processes of the banking service supply chain and formulated the service supply chain process system. The processes of the bank service supply chain, based on the proposed process system, were divided into two categories: (1) operational processes (e.g., planning and formulation of strategies, designing services and products, marketing and the sales of products and services, product delivery, service delivery, and public relations and advertising management), and (2) support and management processes (e.g., financial resources management, human resources management, IT management, risk and credit management, the bank's branch and state administration, marketing and sales management, and business improvement management). The intuitive fuzzy DEMATEL method was used to determine the most effective service supply chain processes. The results indicate that the processes of goods delivery, risk management, IT management, public relations and advertising management, and marketing and sales management had the greatest impact on the processes of the banking service supply chain. Consistent with the results of this section, the effectiveness of the information technology process was also confirmed by [42,48–50]. In addition, the effectiveness of the processes of the marketing and sales management and the public relations and advertising management was also confirmed by [51–55]. The effect of the risk management process on other processes of the banking service supply chain was also confirmed by [56–60].

With the comments provided by the panel, and according to the PCF framework of Tejarat Bank, the effective processes were confirmed and the conceptual model of the research was developed. The use of the IMC program develops the sub-processes of the marketing and sales management, and, by recognizing the characteristics of the target customers in the target markets, the amount of deposit-making in the bank is increased by the target markets; this affects the bank's liquidity risk. It should be noted that in this model, the current activities realized in the platform of information technology and the process of managing goods delivery were integrated with the process of public relations and advertising management. The model, after applying the proposed adjustments, was verified by the consultation group.

During the running of the model, marketing rules were implemented in the simulation space and the results were analyzed by the Taguchi method. The budget and time constraints prevent banks from simultaneously increasing the effectiveness level of all elements of the IMC program to the highest level. Therefore, it was necessary to determine the most desirable combination of the effectiveness level of each of the elements. Therefore, the outputs of the liquidity risk analysis, with respect to two indexes used in calculating the liquidity risk, were used to determine the most desirable combination of elements of the IMC program in the academic target market. The most desirable combination of IMC elements mostly focuses on the allocation of budget and time to increase the effectiveness levels of the elements of direct marketing, digital marketing, and sales promotion, so that

the fourth effectiveness level (i.e., 0.75–1) for these is realized. It is recommended that the bank redesigns its business processes, employs smart devices in working units, and develops its IT infrastructure to move towards digital banking. Furthermore, by understanding the customers' mental structure and by designing websites, they can reinforce the digital marketing space and maximize the effectiveness of this type of marketing tool. In addition, banks provide multiple products to a wide range of customers through multiple communication channels in different time periods; they have difficulties in providing the effectiveness of direct marketing programs.

Therefore, in every target market, the characteristics of the banking customers and their ability and willingness to use the banking services need to be identified and, accordingly, customers need to be classified. In this way, the provision of the right product for the right customer at the right time can be realized. Advertising partnerships, with respect to their share, places in the second level of effectiveness (i.e., 0.25–0.5). The bank can be creative in its advertising designs and develop a model for evaluating the effectiveness of advertising to further increase the impact of advertising on customers. In addition, the bank can implement the model at specific time intervals to examine the effectiveness of advertising. It is also desirable to identify all the bank advertising channels and to redesign and revise the advertising tools based on the characteristics of their partners in the academic target market, including students, employees, and professors. This way, the mental involvement of the customers with the bank's services and facilities will increase and the desired level of effectiveness will be realized.

## 5. Discussion

The model is based on configuring selected processes of the banking service supply chain. The banking service supply chain processes were identified and extracted by using the comprehensiveness feature of the process classification model as well as expert opinions, and the selected processes were identified using a multi-criteria decision-making approach, which was based on inter-process relationships and interactions. In order to confirm the service supply chain model presented by [8,61], which emphasizes the existence of the network and communication structure in the service supply chain, the relationships and interactions of the processes of product delivery, risk management, IT management, public relations and advertising management, and marketing and sales management as the selected processes of the banking service supply chain were explained by comments of the panel members, and in this way, the structural and communication dimension of the bank's service supply chain was estimated. Since the customer's pivotal role as the main characteristic of service was not confirmed in any of the presented models, a set of behavioral rules using the experimental design method was designed for customers and included in the model for further emphasis on the role of the customer in banking service supply chain management. The investigation into the service supply chain models showed that the dynamic nature of services and the flow of existing processes in the service supply chain were not considered. Therefore, the present study implemented a research model using agent-based modeling and simulation to encompass the complexity of the banking service supply chain.

The proposed framework can be implemented with the other target markets of the bank; the IMC program can be designed and explained based on the characteristics and the behavior type of the existing factors, as well as on the way they interact. In addition, bank customers have different views about different banking products in a certain period of time, which, in turn, varies, depending on the entrance of different steps in the life cycle. It is recommended that researchers add the banking product life cycle to the model. This way, depending on the step of the product life cycle in which the product is placed, we can plan the appropriate schedule for implementation as well as the suitable number of repetitions for each of the sales promotion elements.

From a practical point of view, in recent years, new achievements in banking have created a highly competitive market. Improving the current state of the banking industry

requires that supply chain procedures and processes in the banking system be fundamentally reviewed. One of the important issues of the country's banking system is that the executive and management processes are not systematically designed and developed on a comprehensive and integrated framework. Therefore, in order to change the existing conditions, it is necessary to review and optimize the supply chain processes of the banking system so that constructive interaction can be established among the issues that form the processes, and inter-process interactions can be improved. Thus, we can achieve process integration in the banking system. This study was designed to identify the key elements and components of the selected processes in the supply chain of the banking system by focusing on the current and future conditions of the system and by developing scenarios related to each process to explain how the elements and components interact with each other. We hope that the results of this research will assist managers in reviewing and integrating the processes by clarifying how the elements interact in the selected processes of the banking service supply chain. Furthermore, the scenario approach makes it easier for managers to analyze the results of uncertain processes and can be effective in determining the competitive situation of the banking system. As a practical suggestion, it is desirable for the managers of ITB to implement the improvement cycle in the organization according to the most desirable combination of elements of the integrated marketing communication program. The proposed cycle is shown in Figure 7:

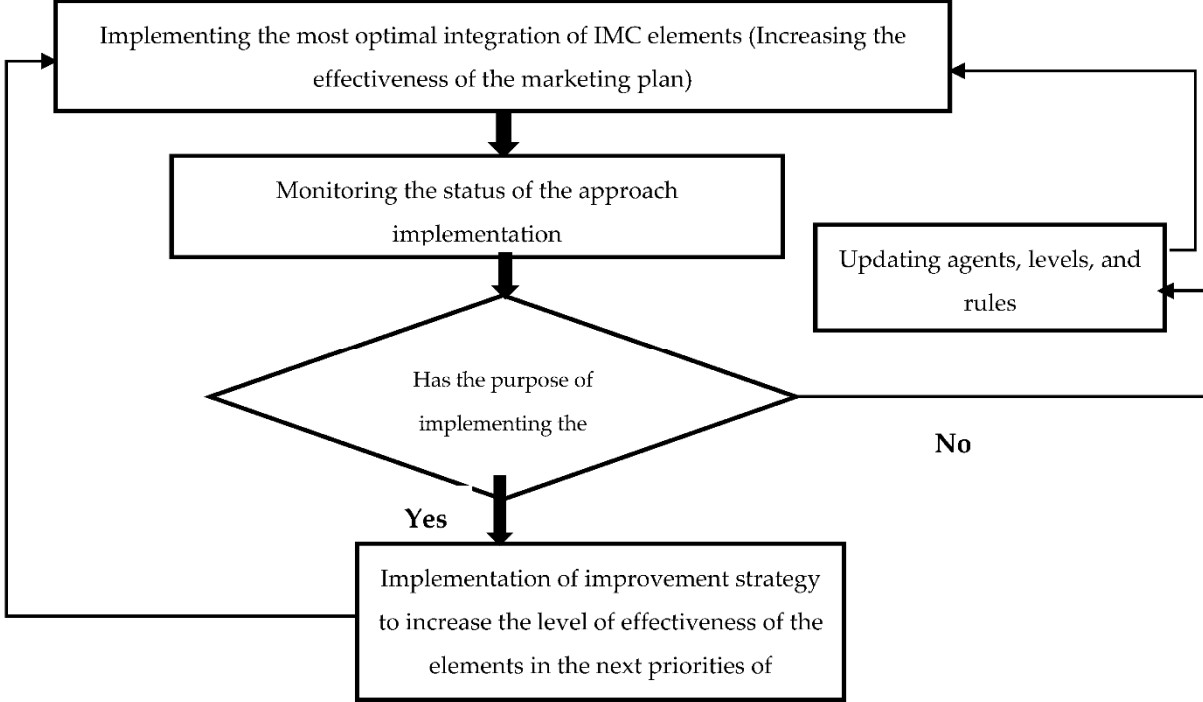

**Figure 7.** Improvement cycle.

Therefore, the most optimal combination of IMC elements will be implemented in a certain period of time, and its ability to improve the effectiveness of marketing programs can be measured. If in practice, the selected combination leads to improvement, future marketing plans should be taken into consideration (as long as its implementation guarantees improvement). Otherwise, changes should be made to the if/then rules, and the can cycle continue to run by explaining the new combination of elements.

In the present study, the researchers were not allowed to access the database containing bank customers' demographic data and the bank deposits, and were thus unable to develop behavioral rules related to depositing and receiving loans and bank facilities. Therefore, the behavioral rules were developed based on the data collected in the panels, limiting the accurate calculation of liquidity risk.



**Author Contributions:** Conceptualization, S.M.Z., N.J. and M.G.; methodology, S.M.Z., N.J., A.N.S. and A.A.; software, A.N.S. and N.J.; validation, S.M.Z., A.N.S. and N.J.; formal analysis, N.J., A.N.S. and S.M.Z.; investigation, N.J. and S.M.Z.; resources, N.J.; data curation, S.M.Z. and A.N.S.; writing— original draft preparation, N.J., S.M.Z. and M.G.; writing—review and editing, N.J., S.M.Z., M.G. and A.A.; visualization, N.J.; supervision, S.M.Z.; project administration, S.M.Z. and N.J.; funding acquisition, S.M.Z. All authors have read and agreed to the published version of the manuscript.

**Funding:** This research was funded by Iran National Science Foundation, grant number 96001832.

**Institutional Review Board Statement:** Not applicable.

**Informed Consent Statement:** Not applicable.

**Data Availability Statement:** Not applicable.

**Conflicts of Interest:** The authors declare no conflict of interest.

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
