# Peer review of "Agent-Based Approach to Configure Processes in Iran’s Banking Service Supply Chain"

_sustainability, doi:10.3390/su13147566_

Round 1

Reviewer 1 Report

The way of presenting the information is relevant and, to some extent, fluent, but in some cases the transition from one section/research part to another is quite sudden, which lead to a decrease in the understanding level and a loss of attention of the audience. It is suggested to address and re-examine the issues mentioned above.

Please pay special attention to the Materials and Methods section. The usage of figures is useful in presenting the research steps. However, the authors could consider clarifying the methodology.

Moreover, the detailed exposure of the research hypotheses (the main hypothesis and, if applicable, the specific hypotheses) is highly recommended. Even if the author/s mentioned the research questions, in the absence of clearly defined hypotheses, there is the possibility of misunderstandings on the part of the audience.

Within the discussion section, the authors managed to summarize and to strengthen the analysis' results. Despite this, the conclusion is not directly exposed, which could easily determine confusion among readers. Therefore, a clear distinction between the discussion section and the conclusions section should be taken into consideration.

The final conclusions should also provide concise viewpoints related to the limitations of the current study. Awareness of the limitations of research is a strong point of it, as it lays the foundations for any possible future work.

Author Response

Dear Managing Editor of the Sustainability journal,

We thank you very much for your time and care about our submission. We also thank the reviewers for their time and for their valuable comments.  The comments are taken into consideration as explained below.

Thank you

With much appreciation

Reviewer 1:

Comments and Suggestions for Authors

The way of presenting the information is relevant and, to some extent, fluent, but in some cases the transition from one section/research part to another is quite sudden, which lead to a decrease in the understanding level and a loss of attention of the audience. It is suggested to address and re-examine the issues mentioned above.

  • Please pay special attention to the Materials and Methods section. The usage of figures is useful in presenting the research steps. However, the authors could consider clarifying the methodology.

Answer: Thank you. According dear reviewer comment, we tried to add a comprehensive description about research steps in Materials and Methods.

The processing system was designed at three steps. At first, the supply chain processes for Tejarat Bank were identified. We referred to the management units in Tejarat Bank, reviewed the job description and upstream documents of the banking units, and interviewed the active people in the management units (at the level of manager, deputy, and expert) who were also the members of the panel. Second, the process framework of Tejarat Bank was set based on the PCF process framework. The extracted processes in the first step were then set in the framework of PCF model for banking processes, so the extracted processes and PCF model processes that were not mentioned in the panel discussions, along with the sub-processes, were formed into a questionnaire. Using the Delphi method, we discussed it again in the panel with the bank experts for review. After the panel, the items that were irrelevant and disproportionate to Tejarat Bank processes were removed and the necessary items were added to the questionnaire so that the process framework for Tejarat Bank could be designed and compiled based on the PCF process framework. Lastly, the final framework was introduced for Tejarat Bank service supply chain processes.

Stage two: Determining the selected processes of Tejarat Bank service supply chain.

The second stage was performed at five steps. Based on the framework of processes designed at the previous stage, a questionnaire was developed to identify the relationships within the service supply chain processes in Tejarat Bank. The questionnaire was then distributed among the respondents. After the data were collected through the questionnaire, the respondents' opinions were transformed into trapezoid intuitive fuzzy linguistic variables. In the next step, the collected data were analyzed by the Intuitionistic Fuzzy DEMATEL Methodology, and the most effective processes of Tejarat Bank service supply chain were introduced. At last, the results of the Intuitionistic Fuzzy DEMATEL analysis were presented to experts for confirmation and finalization.

Stage three: Developing a conceptual research model. First, in order to identify the structure and purpose of each of the selected processes of Tejarat Bank service supply chain, the members of the panel were interviewed, and articles published in reputable scientific databases were read. Content analysis was employed to determine the most important factors affecting the selected processes of Tejarat Bank. In order to extract the most important effective factors, the articles related to the subject were read, and the members of the panel were interviewed. After extracting and coding the factors, we decided on suitable themes for the conceptual categories. The results for this stage was a set of the most important factors affecting the selected processes of Tejarat Bank service supply chain. According to the structure and purpose of the selected processes of Tejarat Bank service supply chain and the result of the content analysis, a conceptual research model was designed. This model was presented to the members of the panel of experts to receive their opinions in order to finalize the conceptual research model and to be used in the fourth step.

Stage four: Agent based modeling cycle. The agent-based modeling cycle included five steps: a) formulating the research question, b) compiling the research hypotheses, 3) determining the structure of the model, 4) applying and implementing the model, and 5) analyzing the model. Modeling began with a clear and precise research question about the conceptual model. In the second step, the related hypotheses were formulated for the processes and structures involved in the conceptual research model. Then the factors involved in the selected supply chain processes were identified and the behavior of each factor and the modeling environment were defined. Some behaviors of the factors involved in the model were explained by the design approach of Taguchi experiments. In the application and implementation step of the model, an attempt was made to define the factors involved in the model in the form of moving objects and the behavior of the factors in the form of mathematical relations and to program them in the Anylogic 8.8.2 software environment.

Stage five: Analyzing the results and reporting the findings. At this stage, the results of the implementation of the scenarios in the fourth stage were analyzed and suggestions were made in order to improve the bank service supply chain processes and to develop the model.

  • Moreover, the detailed exposure of the research hypotheses (the main hypothesis and, if applicable, the specific hypotheses) is highly recommended. Even if the author/s mentioned the research questions, in the absence of clearly defined hypotheses, there is the possibility of misunderstandings on the part of the audience.

Answer: Thanks for the great comment. Since the simulation models aim at explaining the reality in the model, they are always accompanied by research questions. Therefore, in the present study, which is drawn upon agent-based modeling and simulation, a research question was made, and the role of the hypothesis is played by the scenario analysis.

  • Within the discussion section, the authors managed to summarize and to strengthen the analysis' results. Despite this, the conclusion is not directly exposed, which could easily determine confusion among readers. Therefore, a clear distinction between the discussion section and the conclusions section should be taken into consideration.

Answer: Thanks for your great suggestion The Results and Discussion sections were separated by adding the heading "Results" to make it easier for the readers to follow. Thus, in the Result section, the results of data analysis and the answers to research questions were expressed. In the "Discussion" section, we tried to explain the advantages of the model proposed in the research by examining the bank service supply chain models.

  • The final conclusions should also provide concise viewpoints related to the limitations of the current study. Awareness of the limitations of research is a strong point of it, as it lays the foundations for any possible future work.

Answer: Thanks for your consideration. Regarding your point of view, we added research limitation to the last section of manuscript.

In the present study, researchers were not allowed to access the database containing bank customers' demographic data and the bank deposits, being unable to develop behavioral rules related to depositing and receiving loans and bank facilities. Therefore, the behavioral rules were developed based on the data collected in the panels, limiting the accurate calculation of liquidity risk.

Reviewer 2 Report

In my view, this paper contributes to literature. However, the paper may not be accepted unless the following issues are fully addressed.

  1. I recommend the authors to improve the introduction. Sign more clearly what are the contributions they make to the state of art.
  2. It is necessary to specify the aim of the paper explicitly and precisely in the introduction.
  3. The structure of study should be added to introduction.
  4. Authors should improve the theoretical framework briefly.
  5. Conclusion and Policy implication should be added as separate part.

Author Response

Dear Managing Editor of the Sustainability journal,

We thank you very much for your time and care about our submission. We also thank the reviewers for their time and for their valuable comments.  The comments are taken into consideration as explained below.

Thank you

With much appreciation

Reviewer 2:

Comments and Suggestions for Authors

In my view, this paper contributes to literature. However, the paper may not be accepted unless the following issues are fully addressed.

  1. I recommend the authors to improve the introduction. Sign more clearly what are the contributions they make to the state of art.

Answer: So much thanks. Regarding dear reviewer suggestion, we tried to bold our research contribution by adding below descriptions to introduction section.

In recent years, the new achievements in the area of banking, have created a highly competitive market. The improvement in the current situation of the banking industry requires that the procedures and processes of the supply chain of the banking system be fundamentally reviewed. So far, limited studies have been conducted on bank service supply chain, and in none of the proposed models, the customer's central role as the main feature of the service, the processes involved in the bank service supply chain, and the existing interactions and inter-process communications have been taken into account. The contribution of this study is to provide a model for the service supply chain of Tejarat Bank, which was followed by examining the architecture of selected processes of the banking system.

Unlike the methods discussed earlier, agent-based simulation and modeling are used to understand systems that consist of a large number of interacting agents. The agent-based method considers agents and interactions between them at any level of complexity and places them in computational frameworks. The agent-based method is used when in the system, the agents are dynamically connected with each other and participate in the formation of dynamic strategic behaviors in addition to having the power to learn and gain experience. Despite the fact that the agent-based simulation method has superior features over other simulation and modeling methods such as system dynamics, innovative algorithms, etc., and can also be used in production and operations management, only few studies have benefited from this method. Regarding its benefits, it is, therefore, necessary to expand the application of this method in service operations management. Also, the studies that have been done so far on service supply chain are limited and their focus has been mostly on the production service supply chain, while there is a need for in-depth studies on the service supply chain. Therefore, it can be acknowledged that ​​the contribution of present study is from two perspectives. It is hoped that the results of this study can create new insights in service-oriented industries.

  1. It is necessary to specify the aim of the paper explicitly and precisely in the introduction.

Answer: Thank you. According your great suggestion, we specified the aim of the paper in the introduction section by adding below description to the research text.

The purpose of this study was to scrutinize the architecture or so-called configuration of selected service supply chain processes in Iran Tejarat Bank (ITB) system by examining the selected processes of the bank service supply chain, clarifying how the elements interact in each of the selected processes, and using the agent-based simulation method. The purpose of configuring and architecting the selected processes of the service supply chain is to achieve the functional goals of each of the selected processes in the banking system. Also, the purpose of the scenario writing approach is to facilitate the analysis of the results of uncertain processes, which can be helpful in determining the competitive position of the banking system.

  1. The structure of study should be added to introduction.

Answer: Thank you. We added the structure of study description to the end of introduction section.

…… For fulfilling aim of this study, at first, all the processes in the bank service supply chain and the set of activities performed in this industry were identified to obtain an overview of the service delivery process in the target industry. Then, the conceptual research model was developed. Then, according to the purpose of the model, the agent-based modeling cycle started, during which actions such as formulating the research question, collecting relevant hypotheses, determining the structure of the model, using and implementing the model, and model analysis were carried out. Finally, after validating the model, we analyzed the results and presented a report on them.

  1. Authors should improve the theoretical framework briefly.

Answer: We deeply appreciate your consideration. It should be mentioned that We made some innovation in dealing with the problem. The conceptual framework of the research presented on Figure 3 in Research model section is a localization of the issues mentioned in the theoretical section. So basically this framework was not designed for similar banks. For improving theoretical framework we added description as below:

Regarding to conceptual model of present reseach, the service supply chain processes of Tejarat Bank were identified and extracted through the comprehensive classification model for processes and the expert opinion in Tejarat Bank to comply with the process dimension of the service supply chain, which is emphasized in all models. The selected processes of Tejarat Bank service supply chain were identified through a multi-criteria decision-making method, based on inter-process relationships and interactions. For the confirmation of the service supply chain model proposed by Sang and Yu (2009) and Lin et al. (2010), which emphasize the existence of a network and communication structure in the service supply chain, the existing relationships and interactions among the selected processes were explained by the expert opinions of the members of the panel, and the structural and communication dimension of Tejarat Bank service supply chain was obtained. Because the central role of the customer as the main characteristic of the service is not emphasized in any of the proposed models, we devised a set of conduct rules for customers and included them in the model to further emphasize the role of the customer in the management of the supply chain of Tejarat Bank. The study of service supply chain models discussed earlier shows that the dynamic nature of service and the processes in the service supply chain had not been cared for. Thus, in the present study, we implemented a conceptual model and agent-based modeling and simulation to take care of the dynamic nature of Tejarat Bank service supply chain.

  1. Conclusion and Policy implication should be added as separate part.

Answer: To comply with the valuable suggestion made by the esteemed referee, we separated the contents of the final part of the research into "Conclusion", "Discussion" and "Limitations of the research". In the Result section, the results of data analysis and the answers to research questions were expressed. In the "Discussion" section, an attempt was made to explain the advantages of the model presented in the research by examining the bank service supply chain models. The practical results were also expressed in this section.

Reviewer 3 Report

The article has no application part at all. Where are the conclusions? I think it would be good to separate part of the Discussion as Conclusions. 
Better explain the novelty and significance of your findings. Consider providing a deeper synthesis of your results, bring some new theoretical findings with a higher level of generalization.

Author Response

Dear Managing Editor of the Sustainability journal,

We thank you very much for your time and care about our submission. We also thank the reviewers for their time and for their valuable comments.  The comments are taken into consideration as explained below.

Thank you

With much appreciation

Reviewer 3:

Comments and Suggestions for Authors

The article has no application part at all. Where are the conclusions? I think it would be good to separate part of the Discussion as Conclusions.

Better explain the novelty and significance of your findings. Consider providing a deeper synthesis of your results, bring some new theoretical findings with a higher level of generalization.

Answer: Thank you so much. To comply with the valuable suggestion made by the esteemed referee, we separated the contents of the final part of the research into "Conclusion", "Discussion" and "Limitations of the research". In the Result section, the results of data analysis and the answers to research questions were expressed. In the "Discussion" section, an attempt was made to explain the advantages of the model presented in the research by examining the bank service supply chain models. The practical results were also expressed in this section.

From a practical point of view, in recent years, new achievements in banking have created a highly competitive market. Improving the current state of the banking industry requires that supply chain procedures and processes in the banking system be fundamentally reviewed. One of the important issues of the country's banking system is that the executive and management processes are not systematically designed and developed on a comprehensive and integrated framework. Therefore, in order to change the existing conditions, it is necessary to review and optimize the supply chain processes of the banking system so that constructive interaction could be established among the issues that form the processes and inter-process interactions would be improved. Thus, we can achieve process integration in the banking system. This study was then designed to identify the key elements and components of the selected processes in the supply chain of the banking system by focusing on the current and future conditions of the system and by developing scenarios related to each process to explain how the elements and components interact with each other. We hope that the results of this research will assist managers in reviewing the processes and integrate them by clarifying how the elements interact in the selected processes of the banking service supply chain. Furthermore, the scenario approach makes it easier for managers to analyze the results of uncertain processes and can be effective in determining the competitive situation of the banking system. As a practical suggestion, it is desirable for the managers in ITB to implement the improvement cycle in the organization according to the most desirable combination of elements of the integrated marketing communication program. The proposed cycle is shown in Figure (7):

Figure 7. Improvement cycle

Therefore, the most optimal combination of IMC elements is implemented in a certain period of time, and its ability to improve the effectiveness of marketing programs can be measured. If in practice, the selected combination leads to improvement, the future marketing plans should be taken into consideration (as long as its implementation guarantees improvement). Otherwise, changes should be made to if-then rules, and the cycle continues to run by explaining the new combination of elements.
